# Hierarchical Debiasing and Noisy Correction for Cross-domain Video Tube Retrieval

## ABSTRACT

Video Tube Retrieval (VTR) has attracted wide attention in the multi-modal domain, aiming to accurately localize the spatial-temporal tube in videos based on the natural language description. Despite the remarkable progress, existing VTR models trained on a specific domain (source domain) often perform unsatisfactory in another domain (target domain), due to the domain gap. Toward this issue, we introduce the learning strategy, Unsupervised Domain Adaptation, into the VTR task (UDA-VTR), which enables the knowledge transfer from the labeled source domain to the unlabeled target domain without additional manual annotations. An intuitive solution is generating the pseudo labels for the target domain samples with the fully trained source model and fine-tuning the source model on the target domain with pseudo labels. However, the existing domain gap gives rise to two problems for this process: (1) The transfer of model parameters across domains may introduce source domain bias into target-domain features, significantly impacting the feature-based prediction for target domain samples. (2) The pseudo labels tend to identify video tubes that are widely present in the source domain, rather than accurately localizing the correct video tubes specific to the target domain samples. To address the above issues, we propose the unsupervised domain adaptation model via Hierarchical dEbiAsing and noisy correction for cRoss-domain video Tube retrieval (HEART), which contains two characteristic modules: Layered Feature Debiasing (including the adversarial feature alignment and the graph-based alignment) and Pseudo Label Refinement. Extensive experiments prove the effectiveness of our HEART model by significantly surpassing the state-of-the-arts. The code is available [1].

## CCS CONCEPTS

• **Computing methodologies → Computer vision**; **Machine learning**.

## KEYWORDS

Video Tube Retrieval, Unsupervised Domain Adaptation

## 1 INTRODUCTION

**V**ideo **T**ube **R**etrieval (**VTR**) task, which aims to ground the target spatial-temporal tubes according to the described language

[1]https://anonymous.4open.science/r/HEART

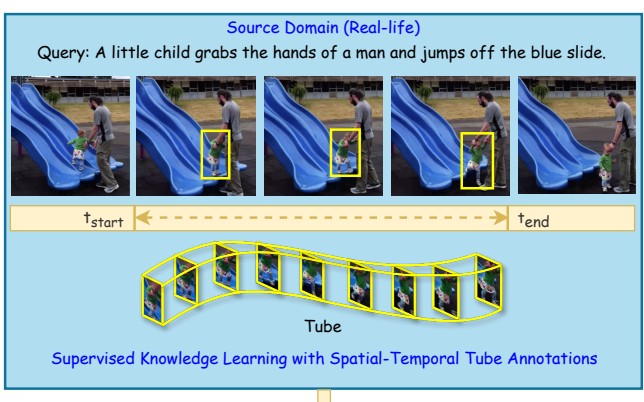

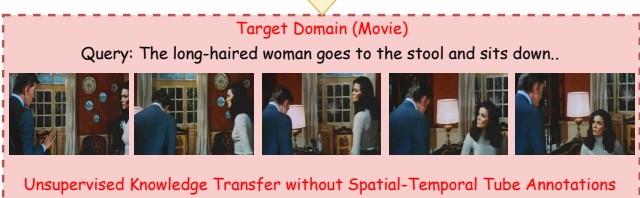

**Figure 1: An illustration of the unsupervised domain adaptation for video tube retrieval (UDA-VTR) task, emphasizing the knowledge transfer from the labeled source domain to the unlabeled target domain without requiring additional annotations.**

sentences, is a fundamental and vital task in the visual-language understanding field [24, 25, 36]. There are mainly two stages in the feature-extracted process for most existing models [24, 36]: **(1) Global Feature Extraction.** Given the video-language pair, the VTR model extracts the cross-modal fusion features (global features) from the multi-modal input. **(2) Local Feature Retrieval.** With the global features, the VTR model further retrieves the local features of the spatial-temporal semantics described by the natural language query for each video frame. Subsequently, the extracted local features undergo further processing through simple linear layers to predict the final spatial-temporal tube. Despite the significant progress, the performance of existing VTR models trained in the specific domain (source domain) drops sharply in another different domain (target domain) due to the domain gap. Intuitively, fine-tuning the VTR models in the target domain with manual annotations may facilitate the model's acquisition of domain-specific knowledge. Unfortunately, it is tremendously time-consuming and labor-intensive to annotate the dense labels of bounding boxes frame-by-frame and temporal boundaries for each video-language sample of the target domain.

To alleviate the unbearable demand for dense annotations of the target domain, we introduce the learning strategy, **U**nsupervised

**D**omain **A**daptation (**UDA**), to the VTR task, which transfers the source domain knowledge to the target domain without the target domain manual annotations. An intuitive solution for the UDA-VTR task is applying the source-domain VTR model (teacher model) to initialize the parameters of the target-domain VTR model, and then generate the pseudo labels of the target domain to fine-tune the target-domain VTR model (student model). However, the presence of the domain gap gives rise to two critical issues that significantly impair the performance of the student model: **(i) Multi-level Feature Bias.** Initializing the student model with the teacher model parameters may introduce the source domain bias from the teacher model into the feature-extraction module of the student model, resulting in the student model producing global and local features influenced by the source domain bias. Due to the domain gap, the source domain bias is not applicable to the target domain, which reduces the precision of the features extracted by the student model, thereby impacting the final prediction accuracy based on those features. *Thus, how to alleviate the source domain bias in both the global and local features extracted by the student model is an essential problem.* Furthermore, the local features capture the visual characteristics of each video frame, which is vital for generating distinct predictions for individual frames. However, such visual characteristics are intertwined with the information of the source domain bias. *Therefore, in addition to eliminating the source domain bias of the global and local features, how to protect each frame's visual characteristics in the local features from being impacted in this elimination process is another problem worth considering.* **(ii) Pseudo Label Noise.** The pseudo labels generated by the source-domain teacher model tend to ground the video clips/video objects widely existing in the source domain samples, which may not be suitable for the target domain samples due to the domain gap. There are mainly two types of failed grounding in pseudo labels: 1) Some pseudo labels fail to capture the essential video semantics corresponding to the natural language query. *Such pseudo labels with excessive deviation may not be suitable for the training of the student model and should be filtered out.* 2) Even if grounding the essential video semantics, there may still be discrepancies in the grounded temporal boundaries of the pseudo labels. As a result, the pseudo labels may exclude the correct bounding boxes of frames within the correct temporal boundaries. This leads to a decrease in the availability of accurate training labels for the student model, which in turn negatively impacts its accuracy. In addition, the pseudo labels may include nonsensical bounding boxes of frames that fall outside the correct temporal boundaries. These nonsensical bounding boxes in the pseudo labels often encompass visual objects not aligned with the natural language queries, and in some cases, even ground background information. *Thus, such pseudo labels with relatively minor discrepancies in the grounded temporal boundaries may significantly disrupt the spatial grounding learning process of the target-domain student model.*

To address the aforementioned issues, we propose the unsupervised domain adaptation model via **H**ierarchical d**E**bi**A**sing and noisy correction for c**R**oss-domain video **T**ube retrieval (**HEART**). Based on the teacher-student framework, our HEART model contains two carefully designed modules: **(i) Layered Feature Debiasing.** To enhance the student model's extraction of global features, we introduce adversarial learning as a training mechanism, which

enables the student model to capture invariant information across different domains. In addition, for the extracted local features, we establish the relation graph to preserve their variance among different frames. Subsequently, we project the local feature graphs of both the source domain and the target domain into the shared eigenspace and align them according to their respective eigenvalues. **(ii) Pseudo Label Refinement.** Firstly, we calculate the confidence of each pseudo label for the target domain samples and filter out uncertain items. Secondly, we extend the grounded temporal boundary in the pseudo labels. Specifically, we identify the frames near the grounded temporal boundary and incorporate the bounding boxes predicted by the source-domain teacher model for these frames into the pseudo labels. Then, we introduce the training weights for the bounding boxes, mitigating the adverse effects of low-quality ones.

Our contributions can be summarized as:

- To the best of our knowledge, we take the early exploration of the unsupervised domain adaptation for the video tube retrieval task. Toward this issue, we propose a novel model, HEART, based on the teacher-student framework.
- We utilize adversarial learning and graph-based alignment to alleviate the source domain bias in the target domain features while retaining the crucial visual characteristics. In addition, we refine the pseudo labels generated for the student model with excessive or minor deviation, respectively.
- Extensive experiments prove the effectiveness of our HEART model by surpassing the state-of-the-arts by a large margin.

## 2 RELATED WORKS

**Video Tube Retrieval.** In the field of video tube retrieval, early techniques [25, 36] typically embraced a two-stage approach. They first necessitated the proposal of potential interest regions through a pre-trained object detector, followed by a custom-designed network to accurately select the relevant regions. The significant limitation inherent in these approaches is their heavy dependence on the capabilities of pre-trained detectors, which inherently constraint their performance. Contrastingly, a paradigm shift is observed in more recent research, [14, 15, 19, 24, 29, 31, 34] have pivoted towards a unified-stage methodology that bypasses the reliance on pre-trained object detectors entirely. For instance, [24] marked a significant milestone by employing a visual-linguistic transformer to achieve simultaneous spatial and temporal localization corresponding to the given textual query. Drawing inspiration from the success of text-driven object detection in [15], [34] unveiled a combined video-text encoder and spatial-temporal transformer decoder to effectively bridge temporal, spatial, and multimodal interactions. In a similar vein, [14] introduced a multi-modal template approach to directly address the issues of alignment and consistency in feature prediction. Furthermore, [19] explores the integration of static and dynamic cues to refine target localization. Despite these advancements, the exploration into how these methods perform when applied to samples that deviate from their initial training set remains largely underexplored.

**Unsupervised Domain Adaption.** Unsupervised domain adaptation, originally applied to image classification, has been broadly adapted for other tasks, among which object detection and video

tasks are most closely related to our work. In recent years, cross-Domain object detection has attracted significant attention from researchers. Adversarial feature learning techniques [7, 23] utilize a domain discriminator to train the feature encoder, facilitating the extraction of domain-invariant features. Meanwhile, image-to-image translation approaches [4, 13] employ generative models like CycleGAN [38] to transform target-domain images into the style of the source domain, thereby bridging the domain gap. Recent efforts in domain adaptation [3, 5, 8, 9, 18, 35, 37] have focused on employing the Mean Teacher [26]. [8] tackles domain shift by leveraging [38] to augment training data, while [18] introduces a combination of weak and strong augmentations alongside adversarial training. Video unsupervised domain adaptation methods typically fall within a few common categories. Adversarial methods [6, 17] align representations across source and target domains through a domain discriminator network. Techniques like DANN [11], originally designed for images, are also applicable to video architectures. [6] is a video-specific approach that uses separate adversaries for the spatial and temporal dimensions. Another class of methods [22] use contrastive learning and exploit the intrinsic structure of video. Despite rapid advancements, to the best of our knowledge, there has not yet been success in applying unsupervised domain adaptation to complex multimodal spatial-temporal video understanding tasks.

## 3 PRELIMINARIES

### 3.1 Adversarial Learning

Adversarial domain adaptation techniques are predicated on the notion of explicitly diminishing the discrepancy between domains by learning transferable features. These methods contrast with traditional supervised learning techniques by integrating a domain discriminator $D$ to differentiate between source and target domain features. Concurrently, the objective of the feature extractor $F$ is to generate features that confuse $D$, thereby promoting learning domain-invariant features. The loss function for the domain discriminator, $L_{\text{dis}}$, is expressed as follows:

$$L_{\text{dis}}(F, D) = \mathbb{E}_{x_i^s \sim D_s} \log[D(F(x_i^s))] + \mathbb{E}_{x_i^t \sim D_t} \log[1 - D(F(x_i^t))], \tag{1}$$

where $D_s$ and $D_t$ represent the features distributions within the source and target domains, respectively. The domain adversarial loss $L_{\text{adv}}$ is formulated using a minimax optimization strategy:

$$L_{\text{adv}}(F, D) = \max_F \min_D L_{\text{dis}}(F, D). \tag{2}$$

To facilitate the minimax optimization, Gradient Reversal Layers (GRL) [10] are inserted to reverse the gradient during backpropagation from $D$.

### 3.2 Graph Spectra Theory

In graph theory, the distance between graph spectra quantifies the divergence in spectral characteristics of graphs with an equal number of vertices, simplifying the comparison of their discrepancies [1, 28]. Drawing from [33], we present the definitions of graph Laplacians and spectral distances.

*3.2.1 Graph Laplacians.* Consider a finite graph $G = (V, E)$, comprising a set of vertices $V$ and weighted edges $E$. Define a vertex function $\varphi : V \to \mathbb{R}$, mapping each vertex to a real value, and an edge weighting function $\gamma : E \to \mathbb{R}$, assigning weights to edges. The graph Laplacian $\Delta$, operating on $\varphi$ for a vertex $v$, is given by

$$(\Delta\varphi)(v) = \sum_{u:d(u,v)=1} \gamma_{uv}(\varphi(u) - \varphi(v)), \tag{3}$$

where $d(u, v)$ denotes the distance between vertices $u$ and $v$, and $\gamma_{uv}$ represents the weight of the edge connecting $u$ and $v$.

*3.2.2 Spectral Distances.* Consider two distinct, nonisomorphic simple graphs $G_s$ and $G_t$, each with $n$ vertices. Let their Laplacian spectra be denoted as $\Lambda_s = \{\lambda_{s_i}\}_{i=1}^n$ and $\Lambda_t = \{\lambda_{t_i}\}_{i=1}^n$, where $\lambda_{s_1} \geq \lambda_{s_2} \geq \ldots \geq \lambda_{s_n}$ and $\lambda_{t_1} \geq \lambda_{t_2} \geq \ldots \geq \lambda_{t_n}$, respectively. The spectral distance $\sigma(G_s, G_t)$ between $G_s$ and $G_t$ is defined by

$$\sigma(G_s, G_t) = \|\Lambda_s - \Lambda_t\|_p, \quad p \geq 1. \tag{4}$$

This metric quantifies the dissimilarity between the graphs based on their Laplacian spectra, facilitating a nuanced comparison of their structural properties.

## 4 METHODOLOGY

Before delving into the details of our proposed methodology, we first outline the problem formulation and revisit the base video tube retrieval model and the teacher-student framework in Section 4.1, which forms the basis of our study. Subsequently, we introduce the Layered Feature Debiasing (LFD) module in Section 4.2 and the Pseudo Label Refinement (PLR) module in Section 4.3. Finally, we describe the overall training pipeline in Section 4.4.

### 4.1 Overview

Given an untrimmed video $V = \{v_t\}_{t=1}^T$ with $T$ frames and a textual query $S = \{s_n\}_{n=1}^N$, the objective of the Video Tube Retrieval (VTR) task is to localize a spatial-temporal video tube $B = \{b_t\}_{t=t_s}^{t_e}$ described by the textual query $S$. Here, $b_t$ denotes the bounding box in the $t$-th frame, and $t_s$ and $t_e$ are the start and end frames of the retrieved tube, respectively. In the unsupervised domain adaptation context, we are given a set of untrimmed videos $V^S$ and language queries $Q^S$ in the source domain, and another set of untrimmed videos $V^T$ and language queries $Q^T$ in the target domain. The video-query pairs are labeled with the spatial-temporal tube $B^S$ in the source domain but not in the target one. Under such circumstances, the main objective is to derive an effective VTR model on the target domain by fully exploiting labeled data in the source domain.

State-of-the-art models for video tube retrieval are primarily based on the encoder-decoder paradigm. For example, STCAT [14] firstly utilizes two feature extractors to obtain both visual and textual features from the video frames and query sentence, respectively. It then models the video-text interactions through a spatial-temporal cross-modal encoder, which introduces a video-level learnable token for the whole video to encode target object semantics, and a frame-level learnable token for each individual frame to represent the frame specific appearance. These tokens are further leveraged to produce a template for the target object by a template generator. Finally, the yielded template is treated as a query and fed into a decoder to aggregate features and predict

**Figure 2: Our model features a teacher-student framework, with both initially trained on the source domain. The teacher generates pseudo labels for weakly augmented video-query pairs in the target domain, while the student learns from strongly augmented pairs across both domains, supervised by ground truth and pseudo labels respectively. The Layered Feature Debiasing enables the student model to capture invariant information across different domains through adversarial learning and achieves cross-domain alignment while preserving inter-frame variance within videos through graph based alignment. The Pseudo Label Refinement filters out uncertain pseudo labels and refines them by temporally extending the grounded temporal boundaries.**

the spatial-temporal tube. In this paper, we employ STCAT as the base video tube retrieval model. However, it is important to note that our module is not specifically designed for STACT, but can be universally applied to models based on the encoder-decoder architecture.

Our teacher-student framework is composed of two architecturally identical models: target-only teacher model and cross-domain student model. The teacher model only takes the weakly-augmented video-query pairs from target domain while the student model takes strongly-augmented video-query pairs from both source and target domain. The student model is learned by standard gradient descent, and the teacher model is updated with the exponential moving average (EMA) of the weights from the student model. Thus, the teacher model can be considered as a temporal ensemble of multiple student models: the weights $\theta'_t$ of the teacher model at time step $t$ are derived from the EMA of the student model's successive weights $\theta_t$:

$$\theta'_t = \alpha \theta'_{t-1} + (1 - \alpha)\theta_t, \tag{5}$$

where $\alpha$ is a smoothing coefficient hyperparameter. Then we use the temporally ensembled teacher model to guide the student model's training in the target domain via pseudo labels.

To generate precise and accurate pseudo labels for target domain video-query pairs, we feed the video-query pairs with weak augmentation to the teacher model and pairs with strong augmentation the student model. Specifically, weak augmentations include

random horizontal flipping and cropping, while strong augmentations additionally comprise random color jittering, grayscaling, and Gaussian blurring. We further utilize the random temporal shift perturbation [32] on the encoded features within the student branch. Concretely, a subset of feature channels is selected at random; half of these are shifted forward in time, while the remaining half are shifted backward in time. In such ways, the teacher model's predictions can be more accurate than those of the student model, allowing the student model to learn from pseudo labels generated by the teacher model.

## 4.2 Layered Feature Debiasing

As stated in Section 1, the feature-extraction process of most current video tube retrieval methods may be summarized into two stages: Global Feature Extraction and Local Feature Retrieval. Initializing the student model with the teacher model parameters may introduce the source domain bias from the teacher model into the feature-extraction module of the student model, resulting in the student model producing global and local features influenced by the source domain bias. To alleviate the source domain bias in the extraction of global features, we introduce Adversarial Feature Alignment (Section 4.2.1), which enables the student model to capture invariant information across different domains. Furthermore, to eliminate the source domain bias in local features while simultaneously protecting each frame's visual characteristics, we employ Graph Based Alignment (Section 4.2.2), which establishes a relation

graph that preserves the local feature variance among different frames and projects these graphs from both the source and target domains into a shared eigenspace for alignment according to their respective eigenvalues.

### 4.2.1 Adversarial Feature Alignment.

The visual encoder $E_{\text{vis}}$ independently extracts visual features from each frame in the video, while the textual encoder $E_{\text{txt}}$ encodes the query sentences into the linguistic representation. We incorporate two frame-level domain discriminators, denoted as $D_{\text{vis}}$ and $D_{\text{txt}}$, immediately after the visual and textual encoders and before these unimodal features are concatenated. For the cross-modal encoder, we employ frame-level domain queries to extract and align spatial features for individual frames. Afterwards, video-level domain queries are utilized to extract and align spatial-temporal features across the entire video. The features corresponding to these domain queries are then classified by two respective domain discriminators [30], represented as $D_{\text{frame}}$ and $D_{\text{video}}$. Actually, we reuse the frame-level and video-level learnable tokens in the STCAT encoder instead of introducing another set of tokens, denoted as $E_{\text{frame}}$ and $E_{\text{video}}$. We define the adversarial feature alignment loss, $L_{\text{afa}}$, as an amalgamation of four components: the domain adversarial losses on visual features ($L_{\text{vis}}$) and textual features ($L_{\text{txt}}$), as well as features corresponding to frame-level domain queries ($L_{\text{frame}}$) and video-level domain queries ($L_{\text{video}}$) within their respective encoders. Formally, the adversarial feature alignment loss is defined as follows:

$$
\begin{aligned}
L_{\text{afa}} = \ & \lambda_{\text{vis}} L_{\text{adv}}(E_{\text{vis}}, D_{\text{vis}}) + \lambda_{\text{txt}} L_{\text{adv}}(E_{\text{txt}}, D_{\text{txt}}) \\
& + \lambda_{\text{frame}} L_{\text{adv}}(E_{\text{frame}}, D_{\text{frame}}) + \lambda_{\text{video}} L_{\text{adv}}(E_{\text{video}}, D_{\text{video}}),
\end{aligned}
\tag{6}
$$

where $L_{\text{adv}}$ is defined in Equation (2), $\lambda_{\text{vis}}$, $\lambda_{\text{txt}}$, $\lambda_{\text{frame}}$ and $\lambda_{\text{video}}$ are hyper-parameters that weight the importance of each loss component in the overall adversarial alignment objective.

### 4.2.2 Graph Based Alignment.

We meticulously construct self-correlation graphs for the source and target videos independently to model the relations between different frames in each video. Specifically, given the cross-modal decoder outputs of the source video and target video, denoted as $O_s$ and $O_t$ respectively, our objective is to construct undirected and weighted graphs $G_s = (V_s, E_s)$ for the source domain videos and $G_t = (V_t, E_t)$ for the target domain videos. In these graphs, each vertex $v_i$ belonging to $V_s$ or $V_t$ is represented by a corresponding output vector $O_i$ in the output sets $O_s$ or $O_t$, respectively. The weighted edge $e_{i,j}$ is formulated as the relation between vertices $v_i$ and $v_j$, quantified using a metric function $s$ between their feature representations $O_i$ and $O_j$, based on the Gaussian similarity metric:

$$
e_{i,j} = s(O_i, O_j) = \exp\left(-\frac{1}{2\sigma^2}\|O_i - O_j\|^2\right),
\tag{7}
$$

where $\sigma$ is the standard deviation of the Gaussian distribution and $\|\cdot\|$ represents the Euclidean norm.

For a simple undirected graph with a finite number of vertices and edges, the definition of graph laplacians (Section 3.2.1) is just identical to the Laplacian matrix. We opt for the random walk Laplacian matrix $M$ using the formula:

$$
M = I - D^{-1}A,
\tag{8}
$$

where $A$ is the adjacency matrix of the graph, $I$ is the identity matrix and $D$ is degree matrix of $A$. With the adjacency matrices $A_s$ and $A_t$ of the source and target domain graphs, we can obtain the Laplacian matrices $M_s$ and $M_t$, and their eigenvalues $\Lambda_s$ and $\Lambda_t$, respectively. Following the definition of spectral distances (Section 3.2.2), we can calculate the spectral distances between source domain graph $G_s$ and target domain graph $G_t$, and thus the graph based alignment loss $L_{\text{gba}}$ is defined using the function $\sigma(\cdot)$ detailed in Equation (4):

$$
L_{\text{gba}} = \sigma(G_s, G_t).
\tag{9}
$$

## 4.3 Pseudo Label Refinement

In this section, we first describe the Threshold Filtering (Section 4.3.1) which calculates the confidence of each pseudo label for the target domain samples and filters out uncertain items. Then, we design the Temporal Extension (Section 4.3.2) to extend the grounded temporal boundary in the pseudo labels and introduce the training weights for the bounding boxes.

### 4.3.1 Threshold Filtering.

The pseudo labels generated by the source-domain teacher model tend to ground the video clips/video objects widely existing in the source domain samples. However, these pseudo labels may not be suitable for the target domain samples due to the domain gap and even fail to capture the essential video semantics corresponding to the natural language query. To address this issue, we filter out a portion of the low-quality pseudo labels based on the confidence threshold. Specifically, for each frame, while predicting a bounding box, we also predict an "actionness" score, which indicates the confidence level of the match between the bounding box area in that frame and the associated text. We calculate the average actionness score for all frames within the predicted temporal boundaries, and only videos with an average score exceeding the threshold are included in the training with their pseudo labels. Formally, we define the filter function $F(\cdot)$ as:

$$
F(A, s, e) =
\begin{cases}
1 & \text{if } \frac{\sum_{t=s}^{e} A_t}{e - s + 1} > \tau \\
0 & \text{otherwise}
\end{cases}
\tag{10}
$$

where $A_t$ represents the actionness score at frame $t$, $s$ and $e$ are the predicted start and end of the temporal boundaries, and $\tau$ is the predefined confidence level above which the video is considered suitable for training inclusion.

### 4.3.2 Temporal Extension.

The VTR model predicts the probability of each frame being the start and end boundaries of the pseudo tube (i.e., the tube-shaped pseudo labels predicted by the teacher model), denoted as $\tau_s$ and $\tau_e$, respectively. The start and end times of the pseudo tube, $t_s$ and $t_e$, are determined by the formula:

$$
(t_s, t_e) = \operatorname*{argmax}_{(s,e):s<e}(\tau_s(s) \times \tau_e(e)).
\tag{11}
$$

This formula selects the maximum from the joint start and end probability distribution $(\tau_s, \tau_e)$, excluding invalid combinations where $t_e \leq t_s$. The pseudo tube $\{b_t\}_{t=t_s}^{t_e}$ is formed from bounding boxes $b_t$ within the selected start and end times $t_s$ and $t_e$. For target domain, even when the essential video semantics are grounded, there may still be discrepancies in the temporal boundaries of the pseudo labels. As a result, the pseudo labels may exclude the correct bounding boxes of frames within the correct temporal boundaries.

This leads to a decrease in the availability of accurate training labels for the student model, which in turn negatively impacts its accuracy. In addition, the pseudo labels may include nonsensical bounding boxes of frames that fall outside the correct temporal boundaries. These nonsensical bounding boxes in the pseudo labels often encompass visual objects not aligned with the natural language queries.

We introduce two strategies to extend the predicted temporal interval and refine the calculation of spatial loss via a frame weighting scheme throughout the entire expanded interval, based on the previously obtained actionness scores $\{A_t\}_{t=1}^T$. First, we employ temporal non-maximum suppression (NMS) on $(\tau_s, \tau_e)$ to find the second-best starting and ending frame pair $t_s'$ and $t_e'$, which satisfies the condition that the temporal Intersection over Union (IoU) with $(t_s, t_e)$ is below threshold $\tau_{iou}$, and the average actionness score exceeds threshold $\tau_{act}$. If such a pair does not exist, the attempt is abandoned. Moreover, we bidirectionally expand a new interval $(t_s'', t_e'')$ around the frame $t_c$ with the highest confidence score (opting for the most central one in cases of multiple candidates) such that the difference between the mean confidence score inside the new interval and outside is maximized:

$$t_s'', t_e'' = \operatorname*{argmax}_{s \le t_c \le e} \left( \frac{\sum_{t=s}^e A_t}{e-s+1} - \frac{\sum_{t=1}^{s-1} A_t + \sum_{t=e+1}^T A_t}{T-(e-s+1)}. \right) \quad (12)$$

The final temporal boundary of pseudo tube, $(t_s^{\text{ext}}, t_e^{\text{ext}})$, is derived by unifying the original interval with these expanded ones:

$$t_s^{\text{ext}} = \min(t_s, t_s', t_s''), \quad t_e^{\text{ext}} = \max(t_e, t_e', t_e''). \quad (13)$$

The extension allows more spatial bounding boxes to participate in training. We use the normalized actionness score as a weighting factor for the spatial loss across the entire extended interval.

### 4.4 Overall Training

The overall objective of our model is formulated as follows:

$$L = L_{\text{sup}} + L_{\text{adv}} + \lambda_{\text{gba}} L_{\text{gba}} + \lambda_{\text{unsup}} L_{\text{unsup}}, \quad (14)$$

where $L_{\text{sup}}$ is the supervised loss in the base VTR model, $L_{\text{adv}}$ is the adversarial feature alignment loss defined in Equation (6), $L_{\text{gba}}$ is the graph based alignment loss defined in Equation (9) and $L_{\text{unsup}}$ is the unsupervised loss that computed on pseudo labels that passed through Equation (10) and were processed by Equation (13). $\lambda_{\text{gba}}$ and $\lambda_{\text{unsup}}$ serve as the weighting hyper-parameters.

Prior to the domain adaptation process, the model undergoes initial training with $L_{\text{sup}}$ using the annotated source domain data. This preliminary phase is designed to equip the model with the source domain knowledge, thereby facilitating the generation of informative pseudo labels. Both the teacher and student models are initialized with parameters acquired from this initial training stage. During the domain adaptation stage, we periodically reset the training of the student model's visual encoder, textual encoder and cross-modal encoder to their states trained only on the source domain, as proposed by [37]. After re-initialization, these retrained modules contain no knowledge of the target domain. Subsequently, the enhanced cross-modal decoder component of the student model, along with the teacher model, promotes convergence towards a more favorable optimum.

## 5 EXPERIMENTS

### 5.1 Datasets

We evaluated our model's performance using two domain pairs: Indoor-to-Outdoor Video Tube Retrieval (I2O-VTR) and Real-to-Movie Video Tube Retrieval (R2M-VTR). The I2O-VTR dataset is derived from the comprehensive VidSTG dataset [36], which consists of 6,924 videos annotated for video tube retrieval task, partitioned into training, validation, and test sets with 5,563, 618, and 743 videos, respectively. To facilitate indoor-to-outdoor domain adaptation studies, the VidSTG dataset was manually segmented into two subsets: indoor and outdoor scenes [2]. The segmentation resulted in 3,386 videos designated as indoor (with a split of 3,386/358/429 for training, validation, and test sets, respectively) and 2,177 videos categorized as outdoor (with a split of 2,177/260/314). The indoor domain encapsulates environments defined by limited spatial extents, while the outdoor domain encompasses scenes characterized by unbounded spatial areas. The R2M-VTR dataset incorporates the HC-STVG dataset [25], consisting of 5,660 videos sourced from cinematic productions, along with the entire VidSTG dataset, which encompasses a broad spectrum of real-life scenarios. The HC-STVG dataset is segmented into training and test sets, containing 4,500 and 1,160 videos, respectively. This combination of datasets from cinematic and real-life sources establishes the groundwork for real-to-movie domain adaptation analysis. We adhere to the original dataset splits provided for both the VidSTG and HC-STVG datasets to maintain consistency and integrity in our evaluation.

### 5.2 Evaluation Metrics

To evaluate video tube retrieval, we follow the previous work [36] and define $vIoU = \frac{1}{|S_u|} \sum_{t \in S_i} IoU(b_t, b_t')$, where $S_u$ and $S_i$ denote the sets of frames at the union and the intersection of the ground truth and the predicted timestamps, respectively. The terms $b_t$ and $b_t'$ represent the ground truth and the predicted bounding boxes at time $t$, respectively. We use $m\_vIoU$ as the metric, which is the average of $vIoU$. We also use $vIoU@R$, which is the proportion of samples whose $vIoU > R$. To isolate the evaluation of temporal localization, we employ the metric $m\_tIoU$ (mean of $tIoU$), which is defined as $tIoU = \frac{|S_i|}{|S_u|}$.

### 5.3 Implementation Details

We configure the batch size at 8 and the learning rates for the visual encoder ResNet-101 [12] and the textual encoder RoBERTa [21] at $10^{-5}$, while setting the learning rate for other modules at $10^{-4}$. The AdamW optimizer is employed for training with a weight decay of $10^{-4}$. Our training spans 10 epochs on the I2O-VTR dataset and drops the learning rate by 0.1 after 8 epochs. For the R2M-VTR dataset, the training duration extends to 90 epochs, with a similar learning rate reduction enacted after 50 epochs. To enhance computational efficiency, the input video was uniformly down-sampled to 64 frames and a resolution of 224. The final object tubes are generated through linear interpolation of the predicted bounding boxes in the sampled frames. We empirically determine the hyper-parameters, setting $\lambda_{\text{unsup}} = 1$, $\lambda_{\text{vis}} = \lambda_{\text{txt}} = 0.3$, $\lambda_{\text{frame}} = \lambda_{\text{video}} = 1$ and $\lambda_{\text{gba}} = 100$ in our experiments.

**Table 1: Comparison with baselines on the I2O-VTR dataset. A higher score denotes superior performance, and we highlight the `best` and `second best` scores.**

| Methods | Declarative Sentences | | | | Interrogative Sentences | | | |
|---|---|---|---|---|---|---|---|---|
| | m_tIoU | m_vIoU | vIoU@0.3 | vIoU@0.5 | m_vIoU | m_tIoU | vIoU@0.3 | vIoU@0.5 |
| TubeDETR [34] | 35.77 | 12.72 | 17.02 | 6.97 | 37.49 | 10.61 | 13.38 | 4.17 |
| STCAT [14] | 37.70 | 14.27 | 19.06 | 8.63 | 37.58 | 11.97 | 15.21 | 5.88 |
| STCAT+MDD [16] | 37.96 | 14.83 | 20.31 | 9.35 | 37.89 | 12.40 | 16.05 | 6.34 |
| STCAT+CST [20] | 38.65 | 16.77 | 23.96 | 11.75 | 38.70 | 13.73 | 18.98 | 7.88 |
| STCAT+Ours | 41.33 | 23.22 | 36.80 | 20.81 | 41.63 | 18.91 | 28.85 | 13.83 |

**Table 2: Ablation studies on the I2O-VTR dataset.**

| LFD | PLR | Declarative Sentences | | | | Interrogative Sentences | | | |
|---|---|---|---|---|---|---|---|---|---|
| | | m_tIoU | m_vIoU | vIoU@0.3 | vIoU@0.5 | m_tIoU | m_vIoU | vIoU@0.3 | vIoU@0.5 |
| | | 37.70 | 14.27 | 19.06 | 8.63 | 37.58 | 11.97 | 15.21 | 5.88 |
| | ✓ | 38.91 | 17.08 | 25.23 | 12.76 | 38.90 | 14.12 | 19.94 | 8.55 |
| ✓ | | 39.39 | 18.32 | 27.67 | 14.44 | 39.48 | 15.10 | 21.94 | 9.60 |
| ✓ | ✓ | 41.33 | 23.22 | 36.80 | 20.81 | 41.63 | 18.91 | 28.85 | 13.83 |

**Table 3: Comparison with baselines on the R2M-VTR dataset. A higher score denotes superior performance, and we highlight the `best` and `second best` scores.**

| Methods | m_tIoU | m_vIoU | vIoU@0.3 | vIoU@0.5 |
|---|---|---|---|---|
| TubeDETR [34] | 31.05 | 11.49 | 10.09 | 2.61 |
| STCAT [14] | 32.46 | 12.57 | 11.99 | 3.28 |
| STCAT+MDD [16] | 32.37 | 12.94 | 12.60 | 3.68 |
| STCAT+CST [20] | 33.11 | 13.91 | 15.05 | 5.05 |
| STCAT+Ours | 35.91 | 25.72 | 42.45 | 21.78 |

**Table 4: Ablation studies on the R2M-VTR dataset.**

| LFD | PLR | m_tIoU | m_vIoU | vIoU@0.3 | vIoU@0.5 |
|---|---|---|---|---|---|
| | | 32.46 | 12.57 | 11.99 | 3.28 |
| | ✓ | 33.63 | 16.61 | 22.45 | 9.33 |
| ✓ | | 34.09 | 18.54 | 26.82 | 11.97 |
| ✓ | ✓ | 35.91 | 25.72 | 42.45 | 21.78 |

## 5.4 Performance Comparison

To enable a thorough evaluation of model efficacy across domains, we augment the capabilities of the state-of-the-art video tube retrieval model, TubeDETR [34] and STCAT [14]. Specifically, we first train the model on the source domain dataset and generate pseudo labels for the target domain dataset, which then serve as supervision for training the model within the target domain dataset. For the remaining experiments, we leverage the STCAT model as our base VTR model. Furthermore, we incorporate advanced domain adaptation strategies into the STCAT model, including MDD [16] and CST [20], in an attempt to address the inherent challenges

associated with UDA-VTR task. The comparison results between our model and baselines are shown in Table 1 and Table 3. Beyond components specific to unsupervised domain adaptation, the configuration across the five models is identical.

Our model significantly outperforms all baselines across two datasets. When examining four methods utilizing the same base video tube retrieval model, our approach boosts the m_vIoU metric from 14.27/14.83/16.77 to 23.22 in the I2O-VTR dataset, and from 12.57/12.94/13.91 to 25.72 in the R2M-VTR dataset. The limitations of the state-of-the-art STCAT model stem from its design, which is not tailored for the unsupervised domain adaptation scenario, leading to a lack of nuanced understanding and adaptation to domain-specific challenges. Traditional domain adaptation strategies, such as MDD and CST, fall short of achieving high performance due to the absence of targeted adaptations for the complex UDA-VTR task. Our approach leverages the teacher-student framework, incorporating specific module enhancements to tackle the challenges of unsupervised domain adaptation in the VTR task, achieving substantial improvements.

## 5.5 Ablation Studies

Our model comprises of two core modules: the Layered Feature Debiasing (LFD) and the Pseudo Label Refinement (PLR). Ablation studies conducted on both LFD and PLR, with results detailed in Table 2 and Table 4 for the I2O-VTR and R2M-VTR datasets respectively, demonstrate that the inclusion of either module significantly improves model performance, highlighting their individual effectiveness. When both modules are employed together, the observed synergistic effect surpasses the performance gains achieved by each module in isolation, emphasizing the indispensability and combined efficiency of the LFD and PLR. This synergy underscores the complementary nature of our modules, which can function together and be progressively optimized in a mutual manner.

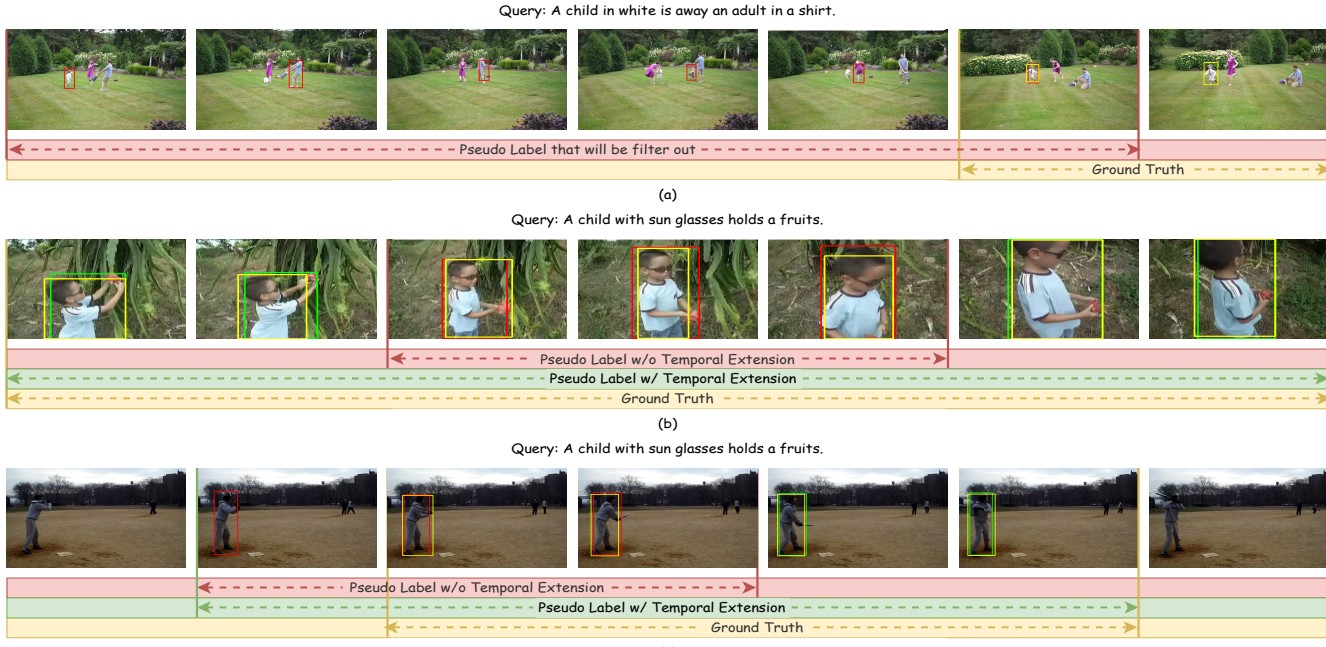

**Figure 3: Visualization examples of pseudo labels that are filtered out and those expanded after filtering.**

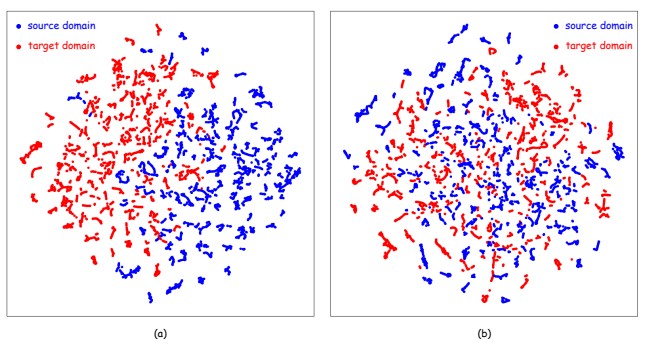

**Figure 4: Visualizations of features with t-SNE comparing before (a) and after (b) applying our method.**

## 5.6 Case Studies

We randomly select several samples from the target domain dataset for case studies and visualize the processes of handling pseudo labels, as shown in Figure 3. From these case studies, we can observe: In case (a), the model fails to capture the essential video semantics that correspond to the natural language query. It predicts bounding boxes that frequently switch among multiple humans, with a lower average confidence level throughout the predicted temporal interval. These predictions will be filtered out by our module and no longer participate in the calculation of unsupervised loss in the current iteration. In cases (b) and (c), the model shows better learning of spatial objects, but its understanding of temporal actions is weaker. This leads to predictions with a higher average confidence

level but with temporal intervals that are either too short or significantly shifted. Our module will temporally extend the pseudo labels in these cases, allowing more spatial boxes to participate in the calculation of unsupervised loss. It is observed that the temporally extended intervals are closer to the ground truth, which aids the model in better learning action information.

We also show the t-SNE [27] visualizations of the features before and after the application of our model in Figure 4. It is evident that there is a significant distribution gap between the source and target domains in Figure 4(a). Our model narrows the domain shift, leading to closely intertwined feature distributions from both domains in Figure 4(b), demonstrating the effectiveness of our approach in mitigating domain discrepancies.

## 6 CONCLUSION

In conclusion, the HEART model significantly advances the field of Video Tube Retrieval (VTR) with its approach to Unsupervised Domain Adaptation (UDA). It effectively resolves major domain adaptation challenges, including source domain bias and inaccuracies in pseudo labels for target domain samples. By integrating the Layered Feature Debiasing module, HEART substantially reduces source domain bias, which enhances the accuracy of target domain feature representations. Concurrently, the Pseudo Label Refinement module elevates the quality and relevance of pseudo labels, ensuring more precise localization of video tubes in the target domain. Our extensive experiments validate that HEART not only outperforms existing state-of-the-art methods and establishes a new benchmark for future research in domain-adaptive VTR, but also highlights the potential of hierarchical debiasing and label correction strategies to effectively tackle the cross-domain challenges.

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
