# OpenReview forum: "Hierarchical Debiasing and Noisy Correction for Cross-domain Video Tube Retrieval"
_acmmm.org/ACMMM/2024/Conference — MM2024 Poster_

### Official Review · Reviewer_Nm9E · 2024-05-11

**Rating:** 4
**Confidence:** 2

**Summary:**

The paper introduces the HEART model to address the domain adaptation challenge in Video Tube Retrieval (VTR) tasks. HEART uses a combination of Layered Feature Debiasing and Pseudo Label Refinement to mitigate source domain bias and enhance the accuracy of pseudo labels used for training in target domains. The results show that HEART outperforms some existing methods, demonstrating its effectiveness in providing more domain-invariant features and reliable pseudo labels.

**Strengths:**

The proposed method utilises adversarial learning to improve the robustness of the model and learn domain-invariant features.

The components used in the model are well-researched and have proven their effectiveness, lending credibility to the proposed method.

**Limitations:**

1. The domain gap between the source and target domains is not significant, raising concerns about the importance of this research topic.

2. The model training involves numerous hyperparameters (e.g., for loss contribution, confidence threshold, etc.) without providing justifications or conducting ablation studies to determine optimal settings. This lack of detailed parameter analysis could hinder the application of this method to other datasets or tasks.

3. The proposed method is complex without an analysis of model size and running time compared with other benchmarks or the baseline method (STCAT).

**Suitability:**

2

---

### Official Review · Reviewer_nKnY · 2024-05-13

**Rating:** 4
**Confidence:** 2

**Summary:**

This paper introduce the unsupervised domain adaptation into video tube retrieval, enabling the knowledge transfer from the labeled source domain to the unlabeled target domain.

**Strengths:**

1. This paper proposed HEART method to solve domain shift problem including layered feature debiasing, pseudo label refinement.
2. This paper exploit UDA for video tube retrieval task via teacher-student framework
3. Adversarial learning and graph alignment are utilized to alleviate the source domain bias

**Limitations:**

1. In table1 and table 3, the author only compare their method with TubeDETR, have they tried other popular method?

2. In Sec.5.3, the author use ResNet-101 and RoBERTa as image and text encoder, respectively, have they tried multi-modal model like CLIP?

3. In the graph establishment, it seems that one frame is treated as a node, so the node number in graph is fixed or dynamic? How does the author deal with this problem.

4. In Eq.(7), the author use Gaussian kernel based distance to generate correlation, have they tried any other distance, e.g. cosine similarity?

**Suitability:**

3

---

### Official Review · Reviewer_MogY · 2024-05-19

**Rating:** 3
**Confidence:** 2

**Summary:**

This paper addresses cross-domain video tube retrieval and develops a HEART approach. The proposed HEART method draws inspiration from the unsupervised domain adaptation and includes two sophisticated modules: (i) Layered Feature Debiasing; and (ii) Pseudo Label Refinement. Furthermore, the HEART is a general strategy that can be incorporated into various encoder-decoder architectures. Extensive experiments verify the effectiveness of the HEART.

**Strengths:**

1. This work takes the early exploration of the unsupervised domain adaptation for the video tube retrieval task, which is an interesting and promising direction.

2. HEART is a general strategy that can be incorporated into various encoder-decoder architectures.

3. The authors open-source their codes, which is good.

**Limitations:**

1. In Figure 2, the abbreviation EMA could be written out as Exponential Moving Average for the sake of readability.

2. In Equation (14), should the term $L\_\{adv\}$ be $L\_\{afa\}$?

3. In the main text, the term $L\_\{unsup\}$  lacks a specific formula, which makes it not easy to understand.

4. The method is constrained by the requirement to configure numerous hyper-parameters, thereby reducing its flexibility. Additionally, the authors have omitted conducting experiments to assess the sensitivity of the parameters.

5. In Equation (10) of the appendix, what are the settings for the additional five hyperparameters introduced here? If $L\_\{sup\}$ is the loss function of STACT [1], I think it is better to cite [1] in the appendix.

6. HEART is a general strategy that can be incorporated into various encoder-decoder architectures. However, in the experiments, the authors solely adopt the STACT architecture. To demonstrate the effectiveness and generality of HEART, it would be beneficial to include experimentation with another architecture.

 [1] Yang Jin, Zehuan Yuan, Yadong Mu, et al. 2022. Embracing consistency: A one-stage approach for spatio-temporal video grounding. Advances in Neural Information Processing Systems 35 (2022), 29192–29204.

**Suitability:**

3

---

### Official Review · Reviewer_C9R9 · 2024-05-25

**Rating:** 5
**Confidence:** 2

**Summary:**

This paper tackles the UDA problem in the VTR task, which enables the knowledge transfer from the labeled source domain to the unlabeled target domain. The porposed method Hierarchical dEbiAsing and noisy correction for cRoss-domain video Tube retrieval (HEART) resolves source domain bias and inaccuracies in pseudo labels for target domain samples.

**Strengths:**

1. The proposed method HEART is well designed and can alleviate the problems in VTR tasks with the UDA settings.
2. The experiments are abundant to support the contributions of this work.

**Limitations:**

1. What was the basis for determining the hyperparameters in Section 5.5, and does it apply to other datasets or scenarios? The discussion about the impact of these hyperparameters would be helpful.
2. The color in Table 1 and 3 can be switched to another format(like texts in bold).

**Suitability:**

3

---

### Meta-Review · Area_Chair_kDaq · 2024-07-03

**Recommendation:** Accept (Poster)
**Confidence:** 5

**Metareview:**

This paper proposes a teacher-student framework to address the unsupervised domain adaptation problem in the Video Tube Retrieval task, which contains a layered feature debiasing module and a pseudo label refinement module. Extensive experiments show the effectiveness of the proposed method. In the rebuttal, the questions and concerns of reviewers have been addressed and all reviewers reach a consensus on the acceptance of this submission.

Quality: It is interesting to explore the unsupervised domain adaptation for the video tube retrieval task. The motivation of this paper is clear and reasonable.

Clarity: The paper is well-written and easy to follow.

Originality: The novelty of this work is sufficient for acceptance.

Significance: The proposed framework has wide applications, especially in scenarios that suffer from the domain gap.